

**Development, characterization and application of an improved online**
**reactive oxygen species analyzer based on MARGA**
Jiyan Wu[1,2], Chi Yang[1,2], Chunyan Zhang [1,2], Fang Cao[1,2], Aiping Wu[1,2], Yanlin Zhang[1,2]
*
[1] *Yale-NUIST Center on Atmospheric Environ., Joint International Research*
*Laboratory of Climate and Environment Change (ILCEC), Nanjing University of*
*Information Science and Technology, Nanjing 210044, China*
[2] *School of Applied Meteorology, Nanjing University of Information Science and*
*Technology, Nanjing 210044, China*
*Correspondence: Yanlin Zhang (zhangyanlin@nuist.edu.cn)*
**Abstract**
Excessive reactive oxygen species (ROS) in the human body is an important factor
leading to diseases. Therefore, research on the content of reactive oxygen species in
atmospheric particles is necessary. In recent years, the online detection technology of
ROS has been developed. However, there are few technical studies on online detection
of ROS based on the DTT method. Here, to modify the instrument, it is added a DTT
experimental module that is protected from light and filled with nitrogen at the end,
based on the Monitor for AeRosols and Gases in ambient Air (MARGA). The
experimental study found that the detection limit of the modified instrument is 0.024
nmol min[-1]. And the accuracy of the online instrument is determined by comparing the
online and offline levels of the samples, which yielded good consistency (slope 0.97,
$R^2$=0.95). It shows that the performance of the instrument is indeed optimized, the
instrument is stable, and the characterization of ROS is accurate. The instrument not
only realizes the online detection conveniently and quickly, but also achieves the hour-
by-hour detection of ROS based on the DTT method. Meanwhile, reactive oxygen and
inorganic ions in atmospheric particles are quantified using the online technique in the
northern suburbs of Nanjing. It is found that the content of ROS during the day is higher
than that at night, especially after it rains, ROS peaks appear in the two time periods of
08:00-10:00 and 16:00-18:00. In addition, examination of the online ROS and water-
soluble ions ($SO_4^{2-}$, $NO_3^-$, $NH_4^+$, $Na^+$, $Ca^{2+}$, $K^+$), BC and polluting gases ($SO_2$, CO, $O_3$,
NO, $NO_x$) measurements revealed that photo-oxidation and secondary formation
processes could be important sources of aerosol ROS. This method breakthrough
enables the quantitative assessment of atmospheric particulate matter ROS at the
diurnal scale, providing an effective tool to study sources and environmental impacts
of ROS.
**1、Introduction**
Air quality is a major issue affecting human health, and prolonged exposure to
high ambient particulate concentrations can lead to a significant increase in the
probability of respiratory and cardiovascular diseases, which can seriously impair
human health (Delfino et al., 2005; Ghio et al., 2012; Pöschl and Shiraiwa, 2015). The
production of reactive oxygen species (ROS) in the human body is the most reliable



pathophysiological mechanism proposed, and excessive reactive oxygen species can
cause an imbalance between the oxidative system and the antioxidant system, causing
oxidative stress and tissue damage (Ahmad et al., 2021; Akhtar et al., 2010; Borm et al.,
2007; Delfino et al., 2013; Lodovici and Bigagli, 2011). Thus, oxidative potential (OP)
has been proposed as a more biologically relevant indicator than particulate matter (PM)
mass concentration to represent the combined effects of multiple toxic components in
PM (Ayres et al., 2008; Hellack et al., 2015; Janssen et al., 2015). Understanding the
generation mechanism and source characteristics of reactive oxygen species is essential
for making reasonable pollution control decisions and reducing their impact on human
health.
In recent years, the analysis method of oxidation potential has cell detection and
cell-free detection. To provide a simpler and quicker way to determine the oxidation
potential of environmental particulate matter, cell-free methods such as electron spin
(or paramagnetic) resonance ($OP_{ESR}$), dithiothreitol assay ($OP_{DTT}$), ascorbic acid assay
($OP_{AA}$), high-performance liquid chromatography (HPLC) and glutathione assay
($OP_{GSH}$) are often used as the main measurement methods for ROS (Bates et al., 2019;
Ghio et al., 2012). Through the comparison and analysis of these various methods by a
large number of researchers, the DTT method is generally considered to be the most
common and comprehensive method to reflect the magnitude of the chemical oxidation
potential of particulate matter (Hedayat et al., 2014; Xiong et al., 2017).
Generally, the cell-free method still has problems with detection delays and
degradation of particulate chemical components during sample storage, which not only
leads to inaccurate detection data, but also the inability to capture daily changes.
Therefore, the development of online detection technology becomes necessary
(Charrier et al., 2016; Dou et al., 2015; Fang et al., 2017; Li et al., 2012; Liu et al., 2014;
Velali et al., 2016; Vreeland et al., 2017). So far, the development of online detection
technology is mainly based on the DCFH method and the DTT method. On the one
hand, an online detection technology based on the DCFH method has been reported
previously (Eiguren-Fernandez et al., 2017; Huang et al., 2016; Sameenoi et al., 2012;
Wragg et al., 2016). However, some researchers believe that in the DCFH method, the
horseradish peroxidase (HRP) will promote the production of hydroxyl free radicals,
leading to an overestimation of ROS content (Pal et al., 2012). On the other hand, based
on the DTT method to develop online detection technology (Fang et al., 2014;
Puthussery et al., 2018), The semi-automatic detection system researched by Fang et al,
based on the DTT method cannot realize an online collection of environmental samples.
On this basis, Puthussery et al used a mist chamber (MC) to continuously collect $PM_{2.5}$
in environmental water and realized fully automatic hourly ROS detection.
However, these detection methods ignore the influence of air and light on the
experiment. As the main reagent of the experiment, dithiothreitol (DTT) and 5,5'-
dithiobis (2-nitrobenzoic acid) (DTNB) are easily oxidized by air (Chen et al., 2010).
Therefore, this experiment is optimized based on the research of Fang et al and
Puthussery et al. We achieve accurate measurement of the oxidation potential of
environmental particulates by shielding from light and filling with nitrogen. In addition,
the present study is developed on the basis of the MARGA, which is a reliable field


instrument. And it is not only used in many research institutes for long-term ion
observation but also used to transform the observation of low-molecular-weight organic
acids in the gas and particle phases (Stieger et al., 2019). MARGA is used to collect
particulate matter and is connected to the optimized $DTT_V$ detection part to observe the
oxidation potential hour by hour. The system realizes simultaneous observation of
oxidation potential and inorganic ions. Here, we optimize the performance of the
instrument and measure the hourly averaged OP of ambient $PM_{2.5}$. The reliability of
online detection of oxidation potential data is supported by analyzing the correlation
between ions, polluting gases, BC and oxidation potential.
**2、 Materials and Method**
**2.1 Instrument set-up and improvement**
Figure 1 shows the scheme and schematic diagram of the system for DTT online
detection. The instrument is set up in the Atmospheric Environ. monitoring laboratory
on the roof of the Wende Building of Nanjing University of Information Engineering
(30 m above the ground) and the room temperature is maintained at 20°C. The entire
system is composed of the MARGA, the automatic sample-receiving device, and the
DTT experimental reaction device. The MARGA is used as an instrument for detecting
atmospheric aerosols and inorganic components of gases (water-soluble ions $Cl^-$、$NO_3^-$、
$SO_4^{2-}$、$NH_4^+$、$Na^+$、$K^+$、$Mg^{2+}$、$Ca^{2+}$), and it collects gases using a wet rotary separator
and aerosols using steam injection, and absorbs gases and aerosols into the aqueous
phase separately to separate them from each other. Then, the resulting solution is
analyzed by ion chromatography equipped with a conductivity detector. That is, the gas
and aerosol are analyzed separately to detect the gas precursors and different ionic
compositions in the aerosol.
In the DTT reaction module, to avoid the influence of light and air on the
experiment, the DTT experimental part in this experiment was kept in an environment
protected from light and flushed into nitrogen. In addition, we added a refrigerator to
store DTT, DTNB and other experimental solutions. During the DTT experiment, the
reaction tube and mixing tube were placed in an incubator at 37°C to simulate the
temperature of human lungs. To realize the subsequent DTT experimental reactions, as
in Figure 1 we collected the liquid-phase aerosols into sample tubes through a dual-
channel split-flow controlled-volume peristaltic pump. And set peristaltic pump 1 speed
to 1.55 ml $h^{-1}$ to finish 1.5 ml $h^{-1}$ sample volume.
Finally, the determination of DTT activity is achieved by the continuous regular
operation of the programmable pumps A and B and the detection of the
spectrophotometer. (see Sect. 2.2.1 for details)
**2.2 Method**
**2.2.1 Online DTT assay measurement**
The whole measurement step is divided into three steps: sample collection, DTT
reaction part, and spectrophotometer detection. In the first step (the sample collection),
the MARGA will discharge 25 ml of aerosol liquid every hour, and use the dual-channel
split flow control volume peristaltic pump 1 to add 1.55 ml of the solution (to ensure





1.5 ml of sample) into the sample tube, and the rest will enter the automatic sampling
device to save through the peristaltic pump 2 (the automatic sampler is set to rotate one
grid per hour).
In the second step (the part is protected from light and in a nitrogen environment),
the reaction part is divided into a DTT oxidation step and a DTT determination
step(Wang et al., 2019). First (DTT oxidation step), use pump A to add 5 mL potassium
phosphate buffer (0.1 mol L$^{-1}$), 1.5 mL aerosol extract sample, and 0.5 mL DTT (1
mmol L$^{-1}$) into the mixing bottle (MV) in sequence. Inhale ultrapure water to clean the
syringe of pump A. DTT reacts with the aerosol extract in MV. Second (DTT
determination step), after completing the first step, immediately use pump A to pump
1mL TCA (10% w/v; quencher) into the reaction flask (RV, wrap it in aluminum foil to
prevent possible Light interference). Then, use pump A to suck the mixed solution in
the 1ml mixing bottle and transfer it to the reaction bottle to mix it with TCA. Add 0.05
mL DTNB (0.01 mmol L$^{-1}$) via pump B and mix. The residual DTT reacts with DTNB
to form light absorption product 2-nitro-5-thiobenzoic acid (TNB) with high extinction
performance at 412 nm.
In the third step, in the detection part of the spectrophotometer, use pump A to add
4 mL Tris buffer (0.4 mol L$^{-1}$, containing 20 mmol L$^{-1}$ EDTA) into the reaction flask
(RV). After the reaction is completed, use pump A to add the final mixture solution in
the reaction flask to the LWCC for the absorbance test. The data acquisition software
(Spectra Suite) records the absorbance at 412 and 700 nm every 10 min (select the
baseline absorbance of TNB). Then, the system uses deionized water (deionized water)
for self-cleaning to eliminate any residual liquid in the reaction flask, tubing, syringe,
and LWCC. To determine the rate of DTT consumption, the time interval is 10 min, and
a total of 6 (0 min, 10 min, 20 min, 30 min, 40 min, 50 min) data points of DTT
concentration over time are generated. Finally, the automated system performs the self-
cleaning procedure again to ensure that there is no residue, and the system repeats the
above operations in the next hour to realize hourly detection of DTT activity.

$$\Delta DTT = -\sigma Abs \cdot \frac{N_0}{Abs_0} \tag{1}$$

$$DTTv = \frac{\Delta DTT_s(nmol\ min^{-1}) - \Delta DTT_b(nmol\ min^{-1})}{V_t(m^3) \times \frac{V_S(mL)}{V_e(mL)}} \tag{2}$$

where σAbs is the slope of absorbance versus time; Abs$_0$ is the initial absorbance
calculated from the intercept of the linear regression of absorbance versus time; and N$_0$
is the initial moles of DTT added in the reaction vial. $\Delta DTT_s$(nmol min$^{-1}$) is the $DTT_v$
consumption rate of the sample, $\Delta DTT_b$(nmol min$^{-1}$) is the blank DTT consumption
rate, $V_t(m^3)$ is the sampling volume corresponding to the sample, and $V_s$(mL) is the
injection volume, $V_e$(mL) is the sampling volume.
**2.2.2 Online DTT instrument performance**
The performance of the automated system is characterized by testing to determine
the instrument response, limit of detection (LOD), precision and accuracy, while using
a large flow sampler to collect samples for offline and online comparative analysis. (See
Sect.3.1 for details)
We perform DTT activity detection and comparison on samples collected by 9,10-





phenanthraquinone(PQN)and traditional high-flow samplers. First, we select PQN
with concentrations of 0.01, 0.02, 0.025, 0.05, 0.085 nmol L$^{-1}$ to compare online and
offline DTT activity detection to determine the error of online and offline experiments.
Secondly, select 10 traditionally collected samples for online and offline comparison,
and then combine the experimental error between online and offline determined by
PQN (PQN online and offline orthogonal fitting) to analyze the accuracy of online and
offline.

### 2.2.3 Instrument maintenance

DTT and DTNB solutions are prepared once every 4 days, and the rest of the
solutions are prepared according to the amount. Before each test, perform an overall
light-proof and nitrogen bag inspection. The standard curve was measured once before
each experiment. PQN is calibrated online and offline ever month. Clean the instrument
pipeline once a week, 5 times each time (Ultra-pure water).

### 2.3 Collection and preparation of environmental samples

The sampling point is located on the roof of the seventh floor of the Maintenance
Branch (34°58′ N, 117°26′ E) of the Power Company, Yunlong District, Xuzhou City.
The surrounding buildings mainly include auto repair shops, logistics centers,
pharmaceutical factories, and large residential areas and farmland. A large flow PM$_{2.5}$
sampler (KC-6120) is used for continuous sampling, and a total of 10 samples are
collected (October 21, 2018-October 31, 2018). When sampling, the flow rate is 1.0 m$^3$
min$^{-1}$, and each sampling time is 24 h; in this study, we collected samples using quartz
filters and stored them in a refrigerator at -26 °C. Before the start of the experiment, the
collected samples were subjected to extraction processing, and a sample film with a
diameter of 16 mm is cut into a brown glass bottle, 5 ml ultrapure water is added to
shake for 30 min, and filtered with a 0.22 μm PTFE syringe filter to remove insoluble
substances.

### 3. Results and discussion

### 3.1 Instrument performance

### 3.1.1 Improvement of the instrument

As we all know, photo-oxidation promotes the generation of ROS (Fang et al.,
2016; Visentin et al., 2016; Yang et al., 2014). In addition, during the measurement
process, the ingress of air inside the instrument will also cause the DTT activity to
increase. Therefore, before on-site deployment, the online DTT inspection instrument
was optimized by filling in nitrogen gas and shielding the whole from light. And
respectively detect the DTT consumption rate (ΔDTT) of 10 blanks (ultra-pure water)
before and after optimization. As shown in Figure 2, before the system optimization,
we found that the average ΔDTT measured by 10 blanks was 0.25±0.04 nmol min$^{-1}$,
and there is a big fluctuation. After optimization, the average ΔDTT is 0.14±0.008 nmol
min$^{-1}$, which is significantly lower than system optimization. Moreover, the standard
deviation (0.008) is much smaller than Puthussery et al (0.08) and Fang et al (0.103)
(Fang et al., 2014; Puthussery et al., 2018). It shows that air and light do promote the
generation of ROS, and the nitrogen environment and avoiding light contribute to the





stability of the system. The optimized system is more accurate in measuring the
oxidation potential of environmental particulate matter. To further prove the
optimization effect, the performance of the instrument is studied. (See Sect.3.1.4 for
details)
**3.1.2 Calibration of DTTv measurement and analysis system**
In past studies, PQN is often used as a standard sample of atmospheric particulate
matter (Charrier and Anastasio, 2011; Charrier and Anastasio, 2015; Xiong et al., 2017).
The analytical measurement part of the online DTT instrument is calibrated by
measuring the DTT activity of PQN at different concentrations. As shown in Figure 3,
the linear graph of DTT consumption rate and PQN concentration, the online detection
slope is 3.66±0.26, and the coefficient $R^2=0.992$. The calibration slope is less than the
slope obtained by the automatic DTT system of Fang et al. (2015) and Puthussery et al
(2018). This also shows that shielding from light and filling with nitrogen will reduce
DTT consumption, and it also supports the accuracy of the system in determining the
oxidation potential of environmental particulates. During the on-site operation, PQN's
online and offline testing is measured at least once a month to ensure online accuracy.
**3.1.3 Limit of detection and precision**
The limit of detection (LOD) of the system is defined as 3 times the standard
deviation of the deionized water blank (N = 23), which is 0.024 nmol min$^{-1}$. To ensure
the accuracy of the system, the deionized water blank samples are taken once a day (14
days) during the sampling period, besides the 10 continuously measured during the
optimization of the system.
Use deionized water to evaluate the accuracy of the environmental sample
automation system and analyze the DTT activity. The low standard deviation
(coefficient of variation, CV=5.61%) of 0.024 nmol min$^{-1}$ indicates that the system has
sufficiently high accuracy for environmental samples.
**3.1.4 Accuracy**
The accuracy of the system is verified by comparing the DTT activity of the
positive control and environmental particulate samples obtained from the automated
method with the results obtained from the same experimental protocol performed
manually. (Cho et al., 2005)
Five concentrations of PQN solutions (0.01, 0.02, 0.025, 0.05, 0.085 nmol L$^{-1}$) are
run in the automatic system, which is very close to the results of the manual system (the
standard deviation of the automatic system is kept at 0.008 nmol min$^{-1}$, and the
coefficient of variation is 2.28 %; the standard of the manual system The difference is
0.0044 nmol min$^{-1}$, the coefficient of variation is 1.48 %). As shown in Figure 4, the
slope (manual/automatic) obtained by orthogonal fitting is 1.14, the intercept is 0.12,
and the correlation coefficient ($R^2$) is 0.997. To ensure the high accuracy of the online
system and the offline system, as a further verification, we used online and offline
manual methods to conduct DTT activity analysis on ten environmental particulate
matter samples.
We use the PQN online and offline DTT consumption rate orthogonal fitting result
as the system to correct the error, as shown in Figure 5, through the offline and online
orthogonal fitting of 10 environmental particulate matter samples before and after the
error correction. We found that the corrected results are better (the slope is 0.97 closer
to 1, the intercept is 0.05 closer to 0, $R^2=0.954$), which is significantly better than the
results of Puthussery et al. The good agreement between the two sampling systems
indicates that the DTT measurement of environmental samples has high overall
accuracy. These tests also proved the necessity of optimization.
**3.2 DTT activity of ambient samples**
The volume-normalized oxidation potential $DTT_V$ is used as an index of exposure
to inhaled air to point out the inherent ability of particles to deplete relevant antioxidants.
During the observation period, the daily change of $DTT_V$ in Nanjing is shown in Figure
6. The average $DTT_V$ is 0.83±0.38 nmol min$^{-1}$ m$^{-3}$. Compared with Beijing's $DTT_V$ in
the spring of 2012 (urban area: 0.24 nmol min$^{-1}$ m$^{-3}$)(Liu et al., 2014; Wang et al., 2019).
and Zhejiang University's annual $DTT_V$ average of 0.62 nmol min$^{-1}$ m$^{-3}$(Yu et al., 2019) ,
our results are on the high side; And compared with Peking University's 2015 annual
$DTT_V$ (12.26±6.82 nmol min$^{-1}$ m$^{-3}$) (Perrone et al., 2016)and Guangzhou's In the winter
of 2017 ($DTT_V$: 4.67±1.06 nmol min$^{-1}$ m$^{-3}$) and in the spring of 2018 ($DTT_V$: 4.45±1.02
nmol min$^{-1}$ m$^{-3}$), our values are low, which may be related to the current season and
emission factors. In addition, we found that the rain during the sampling period caused
significant changes in the 24-hour $DTT_V$. To better understand the environmental
factors affecting $DTT_V$, we divided the $DTT_V$ daily activities. As shown in Figure S2,
the daily distribution of 24-hour DTT activities during the entire sampling period (a),
before rain (b), during rain (c), and after rain (d) are divided. Figure S2(a) represents
the hourly change of $DTT_V$ during the entire sample period. We found that the highest
value of $DTT_V$ in a day occurs at 11-12 am, and $DTT_V$ is greater during the day than at
night, which is similar to the study by Puthussery et al. Before the rain, the average
$DTT_V$ was 0.81±0.17 nmol min$^{-1}$ m$^{-3}$. There is a peak at 10-12 am, but the overall
situation is relatively flat, and there is no obvious difference between day and night.
And the average value of $DTT_V$ during the rain is 0.55±0.10 nmol min$^{-1}$ m$^{-3}$, which
decreased significantly. There is no doubt that this is caused by rain settling the
polluting components of the atmosphere. In contrast, there is significant daily activity
in $DTT_V$ following rain, with peaks occurring mainly between 8-10 am and 4-6 pm, and
$DTT_V$ is significantly higher during the day than at night, which is similar to the
Puthussery study (Puthussery et al., 2018).
**3.3 The correlation between $PM_{2.5}$ and polluting gases and ROS activity**
To further study, the daily changes of $DTT_V$ and its correlation with various
emission sources on site. As shown in Figure 7, we measured the water-soluble ionic
components of $PM_{2.5}$ ($SO_4^{2-}$, $NO_3^-$, $NH_4^+$, $Na^+$, $Ca^{2+}$, $K^+$), BC, and pollution gas ($SO_2$,
$CO$, $O_3$, $NH_3$) content changes. The average concentration of $PM_{2.5}$ during the sampling
period was 9.97±6.53 ng m$^{-3}$, the average concentration of $PM_{2.5}$ before rain was
11.13±7.21 ng m$^{-3}$, the average concentration of $PM_{2.5}$ after rain was 7.80±4.18 ng m$^{-3}$,
$PM_{2.5}$ There is a significant drop in concentration. And through correlation analysis, we
found that $DTT_V$ and $PM_{2.5}$ concentration were positively correlated before rain, but
negatively correlated after rain. Therefore, we suspect that the source of $DTT_V$ is
different before and after the rain. BC and the polluting gases $SO_2$, $NO_x$, $NO_2$, $CO$, $Ca^{2+}$,
$K^+$, $Mg^{2+}$ are often used as tracers of biomass burning, coal combustion, and dust storms.





The levels of these substances were not high during the sampling period and decreased
to varying degrees after rain. It is similar to Liu and Zhang et al who concluded that
biomass burning, coal combustion, and dust storms were not major sources of pollution
in Nanjing during the summer(Guo et al., 2019; Liu et al., 2019; Zhang et al., 2020). In
addition, there was no strong correlation between $DTT_V$ and $SO_2$, $NO_x$, $NO_2$, and CO
before and after the rain. Therefore, it can be judged that neither biomass burning, coal
combustion nor dust is the main source affecting $DTT_V$. In contrast, we found that there
is a significant difference between day and night in $O_3$ after rain, which is similar to the
change of $DTT_V$, and after rain, $DTT_V$ and $O_3$ show a strong correlation (r=0.624). After
it rains, the $O_3$ content in the air environment increases. Under the action of the sun's
ultraviolet rays, the $O_3$ is photodegraded to form active oxygen components such as OH
radicals (Ehhalt and Rohrer, 2000; Rohrer and Berresheim, 2006).
To further confirm the influence of light on $DTT_V$, the day and night correlation
analysis of substances related to photo-oxidation ($NH_4^+$, $NO_3^-$, $SO_4^{2-}$) and $DTT_V$ is
carried out. As shown in Table S2, we find that $NH_4^+$, $NO_3^-$, $SO_4^{2-}$ and $DTT_V$ are
significantly correlated during the day (r=0.434, r=0.461, r=0.263, P<0.01). As far as
we know, there is no evidence in the literature that water-soluble inorganic ions ($NH_4^+$,
$NO_3^-$, $SO_4^{2-}$) have redox activity in an aerobic environment(Calas et al., 2018;
Stevanovic et al., 2017). However, their correlation with $DTT_V$ may be due to
collinearity with redox-active organic compounds, rather than actual contribution to the
oxidation potential of particles. We speculate that the high correlation may be related
to the photochemical reactions that occur during the day.
**4、 Summary and conclusions**
This study proposes and characterizes an improved online active oxygen analyzer.
Compared with the previous research, the main improvements(Fang et al., 2014;
Puthussery et al., 2018). The optimization analysis is as follows: (1) The experimental
environment is processed to isolate the air and avoid light; (2) The sampling method
has changed. We use the MARGA online ion analyzer, which is more mature and stable.
Compared with before optimization, the standard deviation of the blank was
significantly smaller, Thus, the detection limit of the instrument (0.024 nmol $min^{-1}$)
becomes smaller and more stable. The DTT consumption rate is reduced by 24.4 %,
which eliminates the influence of outside air and light in the experiment. And the
consistency between online and offline is improved (slope=0.97, $R^2$=0.95), the
accuracy of the system is higher.
By changing the $DTT_V$ content hour by hour during the sampling period, we found that
the DTT activity during the day is higher than that at night, and it is especially obvious
after rain, which is mainly related to the increase in UV radiation during the day after
rain. In addition, we analyzed the correlation between water-soluble ions ($SO_4^{2-}$, $NO_3^-$,
$NH_4^+$, $Na^+$, $Ca^{2+}$, $K^+$), BC, pollutant gases($SO_2$, CO, $O_3$, NO, $NO_x$, $NH_3$) and $DTT_V$,
and we found that the main source of influence of OP in the Nanjing environment in
summer is daytime Secondary photochemical conversion and ultraviolet radiation. In
the future, we hope to add more experimental modules to the back-end based on the
MARGA sample collection device to realize the diversification of detection
compositions. In addition, the system can be combined with other substance detection



instruments. It will achieve the daily contribution of various emission sources to the
risk associated with OP exposure can be inferred from other species.





*Data availability.* Data used in this paper can be provided upon request by email to
ZYL (dryanlinzhang@outlook.com).
*Author contributions*. WJY designed the instrument, led the sampling campaign,
performed the experiments, and wrote the manuscript. YC participated in experimental
design and guided the experimental process. ZCY chose the building address and
initially built the instrument. CF helped in the filter collection and in conducting the
DTT activity experiments. ZYL conceived the idea, organized the manuscript, and
supervised the project.
*Competing interests*. The authors declare that they have no conflict of interest.
*Acknowledgements*. The authors thank funding support from the National Nature
Science Foundation of China (Nos. 41977305), the Natural Science Foundation of
Jiangsu Province (No. BK20180040), the fund from Jiangsu Innovation &
Entrepreneurship Team.



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

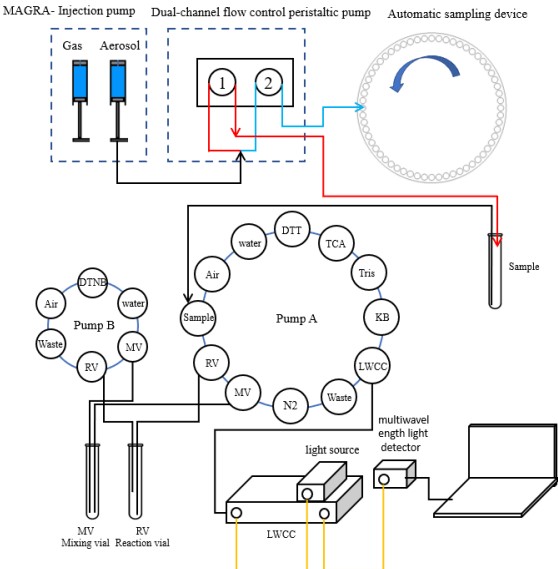


**Figure 1.** Automated system setup (Red line: Peristaltic pump 1 runs at a flow rate of 23 ml h$^{-1}$ for
the first 4 minutes of each hour; Blue line: Peristaltic pump 2 runs at a flow rate of 27 ml h$^{-1}$ for the
remaining 56 minutes of each hour; Yellow line: Optical fiber)

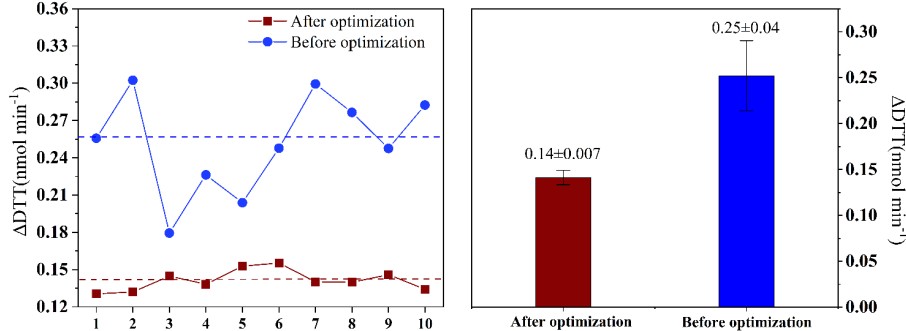


**Figure 2 .**    Comparison of blank DTT consumption rate and standard deviation after system
optimization (the dotted line is the average value)



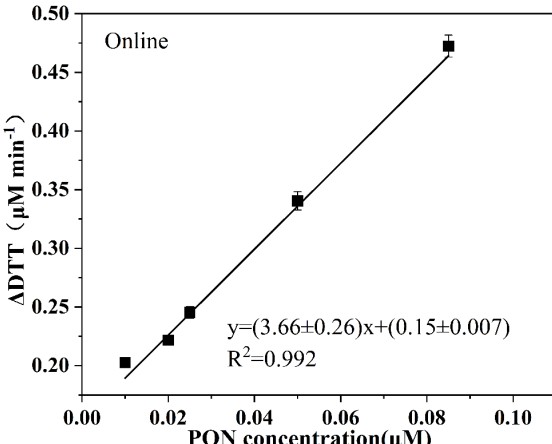


**Figure 3.** Blank corrected DTT consumption rate as a function of PQN used as a positive control.
Each error bar represents the standard deviation of three independent DTT measurements on each
concentration.

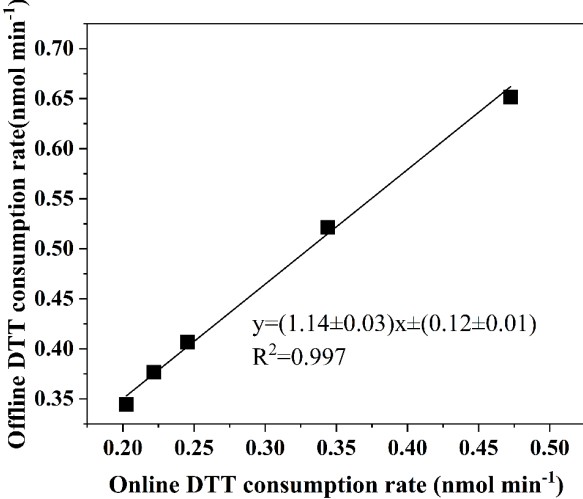


**Figure 4.** Comparison of the automated system with manual operation using PQN (9,10-
phenanthraquinone)

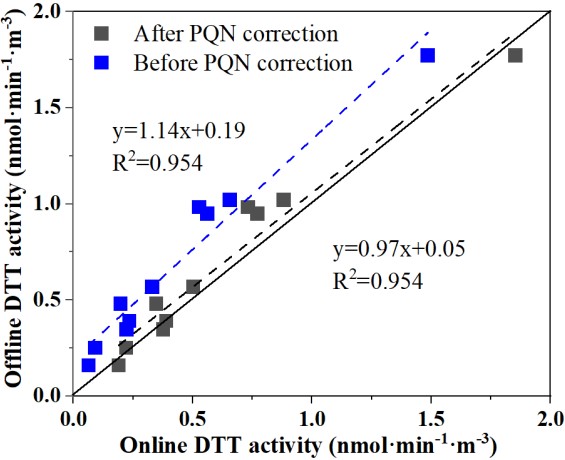

**Figure 5.** Comparison of the automated system with manual operation using ambient aerosol extracts (PM$_{2.5}$ samples collected from Xuzhou, regression analysis is done by orthogonal regression; the line is 1:1).

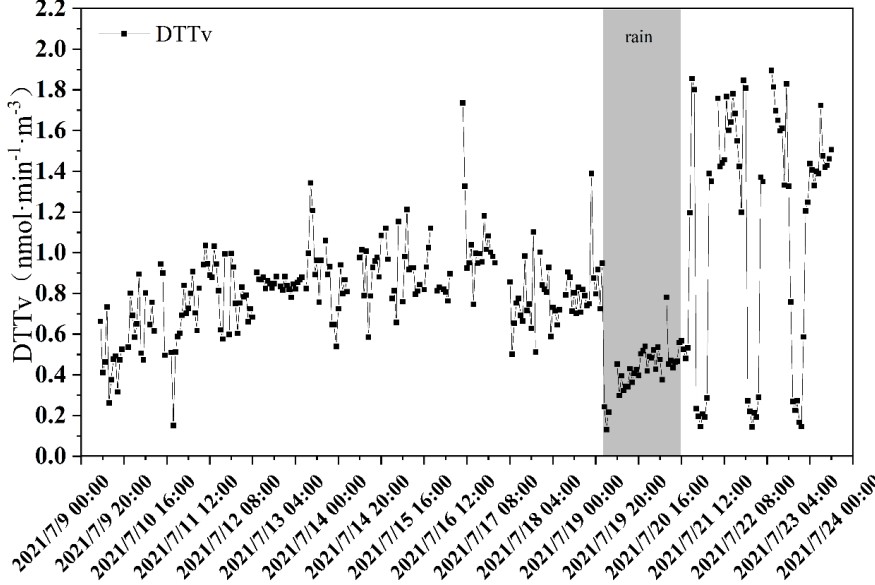

**Figure 6.** Time-series plot of the DTT activity, the shaded part in the figure is the measurement of DTT activity under heavy rain.





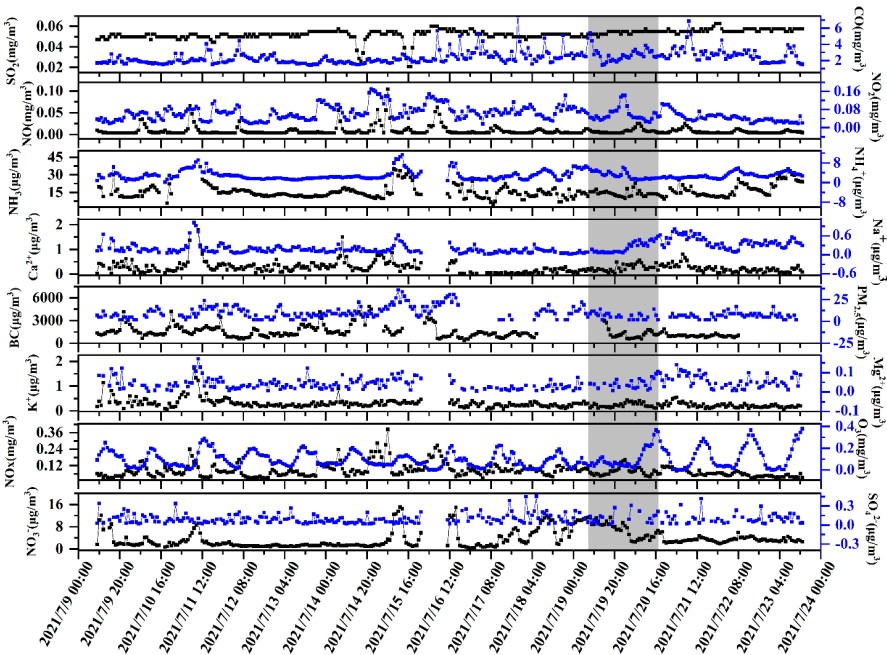

**Figure 7.** Time series of PM$_{2.5}$ water-soluble components (SO$_4^{2-}$, NO$_3^-$, NH$_4^+$, Na$^+$, Ca$^{2+}$, K$^+$) and polluting gases (SO$_2$, CO, O$_3$, NH$_3$) (The shaded part is rainy weather)

**Table 1.** The correlation coefficient (R) between the concentration of water-soluble chemical substances in environmental PM$_{2.5}$ (μg m$^{-3}$) and the volume normalized substance concentration (DTT$_V$), before rain, during rain, and after rain.

| Parameter | Total | Before it rains | During rain | After rain |
|---|---|---|---|---|
| PM$_{2.5}$ | 0.014 | 0.305** | 0.026 | -0.290* |
| SO$_2$ | 0.195** | 0.114 | -0.136 | 0.222 |
| NO | -0.029 | -0.029 | -0.074 | 0.050 |
| NO$_2$ | -0.098 | 0.115 | 0.169 | -0.203 |
| NO$_x$ | -0.085 | 0.062 | 0.142 | -0.169 |
| CO | -0.033 | 0.146* | -0.093 | 0.121 |
| O$_3$ | 0.227* | 0.153 | 0.044 | 0.624** |
| BC | -0.052 | -0.054 | -0.439* | 0.087 |
| NH$_3$ | 0.241** | 0.074 | -0.129 | 0.269* |
| SO$_4^{2-}$ | -0.06 | -0.065 | 0.329 | 0.028 |
| NO$_3^-$ | -0.163* | -0.155* | -0.352* | 0.511** |



| | | | | |
|---|---|---|---|---|
| **NH₄⁺** | 0.024 | 0.028 | 0.062 | 0.271* |
| **K⁺** | -0.077 | -0.045 | 0.125 | -0.337** |
| **Mg²⁺** | 0.131* | 0.075 | 0.233 | 0.086 |
| **Ca²⁺** | 0.005 | 0.072 | 0.021 | -0.055 |
| **Na⁺** | 0.177** | -0.007 | 0.133 | 0.008 |

550    $PM_{2.5}$, particulate matter with an aerodynamic diameter $< 2.5 \mu m$; $*P<0.05$, $**P<0.01$.

551