# Peer review of "Development, characterization and application of an improved online"

_Atmospheric Measurement Techniques, 2021_

## Referee Comment (RC1)

Jiyan Wu et al. present an improvement of the MARGA (Monitor for AeRosols and Gases in ambient Air) instrument by adding a DTT module thus allowing for a simultaneous measurement of some chemical species present in ambient PM as well as gaseous pollutants. They have provided some interesting results on the effect of light and oxygen environments on DTT activity in online DTT instruments. They have also shown some important data regarding the influence (or lack of) of inorganic ions on DTT activity. I believe this manuscript has some good and important results and also provides an inspiration for future modification of traditional aerosol chemical speciation instruments to add a ROS measuring module which might help in a large-scale comparisons of ROS activity of PM and correlation with their chemical composition. I recommend that this paper be accepted for publication in the journal after incorporating some minor comments.

Following are the minor comments that need to be addressed before publication:

- Section 2.1 Please add references for MARGA instrument. Also, a brief description of past studies involving the instrument and what exactly inspired the authors to use MARGA for this study could be added to help the reader better understand the context of the paper.
- Section 2.2.1 It would be a good idea to add a flow chart showing the step-by-step operation of the instrument. Would be helpful for the reader to understand the instrument setup and operation
- 3. Section 2.2.3 Is the instrument maintenance being done manually? Or is it automated? If it is automated would be better to add the steps in the flow chart as suggested in Point 2
- 4. Section 3.1.1 The authors show an improvement of the instrument over other online instruments by comparing how the slope is lower than that of Puthussery et al. and Fang et al. However, it would have been better if the authors could show their own data about experiments conducted side by side (with and without photooxidation). I suggest authors to add this data to further strengthen their point.
- 5. Section 3.1.2 Similarly for this section too, the authors can show their own data to show the effects of photooxidation
- 6. The results from Point 4 and 5 suggested here should be included in the abstract

---

## Referee Comment (RC3)

The author's present a methodology where they incorporate a dithiothreitol (DTT) based assay into a previously established online method for particle composition analysis (MARGA). The authors make alterations compared to previous online DTT methods in the literature, including the use of nitrogen carrier gas and shielding from light in order to reduce the DTT background signal. The optimised DTT method is compared to current methods existing in the literature and deployed in ambient measurement campaigns where they correlate DTT activity with a range of inorganic ions, trace gasses and black carbon. However, there is a lack of technical detail in places, and the comparison between online and offline measurements requires additional clarification. I would recommend publication after addressing the following comments:

Line 111-113 - There is a lack of technical details in general in this section – is nitrogen continuously flowed through, if so what flow rate? How is the DTT reaction vial actually protected from light? As these are key modifications, there should be substantially more technical details added.

Line 172 – Additional details should be given regarding how offline PQN experiments were performed.

Line 174 – What are traditional samples, please specify.

Line 184 – What are the PM2.5 mass loadings on the filters collected for analysis?

Line 204 – Again, more detail required here regarding the nitrogen and light-reducing modifications.

Line 223 – If the slope is lower compared to Puthussery et al, does this not mean the response as a function of PQN concentrations is less, and thus the method is less sensitive to PQN? Is this slope corrected for baseline, accounting for background DTT consumption? This should be elaborated in more detail in the manuscript.

Line 231 – How does this LOD compare to the other methods mentioned in the literature? And how does this compare to the LOD of the offline method used?

Line 254 – It is unclear to me what the purpose of the "PQN correction" is? Please elaborate.

Line 257 – Referring to the comparison of online and offline measurements in Figure 5. The authors state in the introduction that online methods are advantageous due to the online method capturing reactive components that offline methods currently do not, which is valid. Therefore, we would expect the online DTT signal to be higher than that of the offline method for an equivalent sample once normalised, due to the rapid capture of particles in the online method compared to offline. This is not the case in Figure 4 where the PQN slopes are the same, but the offline values are higher compared to online, why? Is this due to the optimisation of the online method? It is not clarified clearly.  In Figure 5, the ambient samples measured offline have a higher DTT activity compared to online, and after the "PQN correction", the offline DTT activity is roughly equivalent to the online, if not still slightly higher for some samples. Puthuserry et al (2018), frequently cited in this manuscript, for instance show a higher online signal compared to offline. There is really limited description of the data in Figure 4 and Figure 5 in the manuscript, and as the online vs offline comparison is a key feature of implementing

the DTT assay into an online methodology, this should be explained more clearly and in more detail.

Line 522 – Figure caption 1 is not sufficient to describe the method, please expand substantially to include more technical detail.

Technical corrections:

The manuscript could benefit from an additional proof reading for English as there are confusing and, in some cases, incomplete sentences throughout the manuscript.

Line 26 – reactive oxygen (species?) typo?

Line 150 – deionized water (deionised water) typo.

Line 180 – the amount of what?

Line 294-296 – ng m-3, typo? ug m-3?

Line 526 – no x-axis title on Figure 2 (left), label both graphs (e.g. A and B)

Line 533 – error given in Figure 3 but not in Figure 4 for the same data?

---

## Author Response (AR1)

**Response to reviewers' comments**

Thank you and the reviewers for handling the manuscript (manuscript number: amt-2021-407). Responses to reviewers are in *italics*. The changes in the manuscript have been marked in blue. Please refer to the point-by-point response to the reviewers' comments and concerns.

Thank you again and the reviewers for such detailed suggestions for revision.

**Comment on amt-2021-407**

**Anonymous Referee #1**

Jiyan Wu et al. present an improvement of the MARGA (Monitor for AeRosols and Gases in ambient Air) instrument by adding a DTT module thus allowing for a simultaneous measurement of some chemical species present in ambient PM as well as gaseous pollutants. They have provided some interesting results on the effect of light and oxygen environments on DTT activity in online DTT instruments. They have also shown some important data regarding the influence (or lack of) of inorganic ions on DTT activity. I believe this manuscript has some good and important results and also provides an inspiration for future modification of traditional aerosol chemical speciation instruments to add a ROS measuring module which might help in large scale comparisons of ROS activity of PM and correlation with their chemical composition. I recommend that this paper be accepted for publication in the journal after incorporating some minor comments.

Following are the minor comments that need to be addressed before publication:

1. Section 2.1 – Please add references for MARGA instrument. Also, a brief description of past studies involving the instrument and what exactly inspired the authors to use MARGA for this study could be added to help the reader better understand the context of the paper.

    **Response:** *Thanks for your comments, we have added to section 2.1 a brief description of past studies involving this instrument and the exact reason for using MARGA in this study. We add a new text to line 110 in the revised manuscript:*

    "In past studies, MARGA was often used to detect the content of inorganic components of atmospheric aerosols and gases in cities around the world (Rumsey et al., 2014). And Chen et al. conducted a special evaluation study on the accuracy and precision of MARGA (Chen et al., 2017). In addition, Stieger et al. achieved quantitative analysis of low molecular weight organic acids in the atmospheric gas phase and particle phase by modifying MARGA (Stieger et al., 2016). Hemmilä et al used a MARGA ligation an electrospray ionization quadrupole mass spectrometer (MS) to achieve 1-hour resolution quantification of 7 different amines in gas and particulate phases in forest air in northern Finland. (Hemmilä et al., 2018)

    As a mature commercial instrument, MARGA can measure the inorganic components of atmospheric aerosols and gases with 1-hour resolution. In this study, based on MARGA, the DTT experimental part is connected to realize the hour-by-hour simultaneous detection of aerosol inorganic components and ROS."

2. Section 2.2.1 – It would be a good idea to add a flow chart showing the step-by-step operation of the instrument. Would be helpful for the reader to understand the instrument setup and operation

    **Response**: *Thanks for your suggestion, I have added a flow chart.*

[Figure]

Figure.2 Schematic diagram of DTT reaction part. (①-④ represents the DTT oxidation step,⑤-⑨ represents the DTT determination step. Blue indicates the ventilation line, all pipelines are wrapped in aluminum foil to protect from light.)

3. Section 2.2.3 – Is the instrument maintenance being done manually? Or is it automated? If it is automated would be better to add the steps in the flow chart as suggested in Point 2

**Response:** *Thanks for your question, the instrument maintenance is done manually, the instrument maintenance is divided into two parts: the maintenance of the MARGA and the maintenance of the DTT experimental part. We add a new text to line 192 in the revised manuscript:*

"The MARGA is calibrated using internal and external standards. The internal standard is a 10 mg L$^{-1}$LiBr solution. The external standard calibration is performed after replacing the anion and cation columns, and the replacement cycle is generally 4 to 5 months. At the same time, the MARGA system is cleaned with 1% hydrogen peroxide and 10% acetone solution, and the airflow is calibrated every two months. In the DTT experimental module, DTT and DTNB solutions are prepared every 4 days. Before each test, perform a comprehensive light and nitrogen bag inspection. To ensure the accuracy of the experimental data, a standard curve is measured before each experiment. The instrument pipeline is cleaned once a week, as shown in Figure 1. The programmable pump A and pump B are connected to the ultrapure water channel. During the cleaning process, all pipelines, reaction tubes and mixing tubes are cleaned."

4. Section 3.1.1 – The authors show an improvement of the instrument over other online instruments by comparing how the slope is lower than that of Puthussery et al. and Fang et al. However, it would have been better if the authors could show

their own data about experiments conducted side by side (with and without photo oxidation). I suggest authors to add this data to further strengthen their point.

   **Response**: *Thanks for your question. This experiment is mainly to compare the changes of DTT consumption rate before and after illumination and nitrogen filling. The slopes of this experiment are simply compared with Puthussery et al. and Fang et al., which are not used as the main basis. In Section 3.1.1, I compared before and after dark and nitrogen charging operations. As shown in Fig. 3, we found that the consumption rate of DTT after shielding from light and filling with nitrogen was lower than that before shielding and filling with nitrogen. The DTT consumption rate of blank sample (ultra-pure water) is reduced by 44 %, which eliminates the influence of outside air and light in the experiment.*

5. Section 3.1.2 – Similarly for this section too, the authors can show their own data to show the effects of photo oxidation.

   **Response**: *In Section 3.1.2, we mainly analyze the calibration of the system, the analytical measurement part of the online DTT instrument is calibrated by measuring the DTT activity of PQN at different concentrations. In Section 3.1.1, I compared before and after dark and nitrogen charging operations. As shown in Figure. 3, we found that the consumption rate of DTT after shielding from light and filling with nitrogen was lower than that before shielding and filling with nitrogen. The DTT consumption rate of blank sample (ultra-pure water) is reduced by 44 %, which eliminates the influence of outside air and light in the experiment.*

6. The results from Point 4 and 5 suggested here should be included in the abstract.

   **Response**: *Thanks for your comments. We have added a new text to line 21 in the revised manuscript:*

   "The DTT consumption rate of the blank sample (ultra-pure water) is reduced by 44 %, which eliminates the influence of outside air and light in the experiment."

**Comment on amt-2021-407**
**Anonymous Referee #2**

Wu et al. presented a work on an optimization of online DTT by adding a DTT experimental module to an online sampler. The authors then compared their online system with manual DTT. The comparison yielded a slope not equal to 1 and the authors used it to calibrate their online data. The authors then measured online DTT from ambient air and compared to water-soluble ions, BC, and gases to found that photo-oxidation and secondary formation processes are important sources of DTT.

There are two major issues with the DTT protocol which will likely lead to large differences in the final DTT activities. The discussion on results and the proofreading need more work.

Therefore, in my opinion, this work needs major revision.

Major comments:

The authors compared the calibration slope from PQN in this work with those from Fang et al. (2015) and Puthussery et al. (2018) and found that their slope is less than the slopes in these two previous studies. They then concluded that "shielding from light and filling with nitrogen will reduce DTT consumption, and it also supports the accuracy of the system in determining the oxidation potential of environmental particulates". The authors also compare the DTT obtained from their system to manual method from Cho et al. (2005), correlation scatter plot shows a slope of 1.14 (off-line methods 14% higher than online). The slope of offline DTT vs online DTT for PM2.5 sample also yielded a slope of 1.14.

All the above seem to suggest that there is a systematic deviation of the online method from this study. The initial DTT concentration used in this work is ~71 micoM (1mM x 0.5mL/7mL). Many other studies used 100uM of initial DTT concentration. In fact, the three references (Fang et al. (2015), Puthussery et al. (2018) & Cho et al. (2005)) the authors used for online-manual comparison all used 100 microM. Since initial DTT concentration makes a difference to the final DTT activity, data obtained in this work is not directly comparable to these three studies and any other studies that use a different initial DTT concentration. The authors need to justify why they used a different concentration.

**Response:** *Thanks for the question, the initial DTT concentration was not fixed, it was determined by the content of the sample (Ayres et al., 2008). This is related to the main principle of the DTT method, which is mainly divided into two parts. In the first part, when DTT is used to measure ROS in atmospheric particulates, reactive oxygen species in particulates oxidize DTT to DTT-disulfide compounds. In the second part, the reaction continues and DTT is continuously consumed. Add DTNB regularly and measure the absorbance of the solution to obtain the remaining DTT concentration. The consumption rate of DTT by particulate matter was calculated (Wang et al., 2019). Therefore, the amount of DTT is related to the sample concentration. The rate of the catalytic redox reaction can be simplified as a straight line (Sauvain et al., 2012), and the correlation of the linear regression reaches more than 0.95.*

In Fang et al. (2015) and Puthussery et al. (2018), the DTT consumption were blank corrected, which means the background activity of light and oxygen should be accounted for already with blank correction. Therefore, the conclusion "shielding from light and filling with nitrogen will reduce DTT consumption, and it also supports the accuracy of the system in determining the oxidation potential of environmental particulates" is not true.

**Response:** *Thanks for your suggestion, we agree with you. Here we mainly reduce the instrument blank. We have revised the manuscript:*

"As shown in Figure 4, the linear graph of DTT consumption rate and PQN concentration, the online detection slope is 3.66±0.26, and the coefficient $R^2$=0.992. During the on-site operation, PQN's online and offline testing is measured at least once a month to ensure online accuracy."

In this work, DTNB and Tris buffer were added to the TCA-DTT mixture, and the absorbance was measured every 10min to get the DTT consumption rates. However, this is wrong, at least speaking from "standard" DTT protocol. DTT consumption should be done in the presence of DTT, buffer, and sample only. This is to make sure the DTT consumption happen at pH 7.4. The correct way is withdraw the mixture of DTT-sample every 10min, then add (TCA), DTNB, and Tris buffer. The authors need to justify why they modify the Cho/Fang/Puthussery protocol while they claim that they are optimizing the DTT assay based on these studies.

**Response:** *Thank you for your question, and I agree with the reviewer. Due to ambiguity in my expression, I rewrite. We add a new text to line 151 in the revised manuscript:*

"Second (DTT determination step), after completing the first step, at 0.10.20.30.40 minutes, use pump A to draw 1ml mixed solution in the mixing bottle and add it to the reaction bottle. Then, immediately add 1 mL TCA (10% w/v; quencher) to the reaction vial (RV, wrapped in aluminum foil to prevent possible light interference) using pump A."

It is not clear to me what new science was obtained from this online system compared to a filter-based system. The importance of contribution of photochemistry and secondary processing to DTT is well studied in tons of previous studies. An online system has a better time resolution compared to a filter based, which is very novel but what this work have found is exceptional is not clear.

**Response**: *Thanks for your question, as you said this study has better temporal resolution. In addition, this study realized the simultaneous detection of inorganic ions and trace gases (water-soluble ions $Cl^-$、$NO_3^-$、$SO_4^{2-}$、$NH_4^+$、$Na^+$、$K^+$、$Mg^{2+}$、$Ca^{2+}$) and $DTT_V$ based on MARGA. Therefore, it is possible to further determine the source of the influence of $DTT_V$. For new scientific discoveries, we need more applications to discover.*

"And through correlation analysis, we found that $DTT_V$ and $PM_{2.5}$ concentration were positively correlated before rain, but negatively correlated after rain." I wouldn't

say for a R value of ~0.3 or -0.2, ie, $r^2$ of 0.09 and 0.04, there is a correlation. This sentence is not statistically supported.

**Response:** *Thanks for your suggestion, we agree with you. I have made corrections. We add a new text to line 318 in the revised manuscript:*

"The average concentration of $PM_{2.5}$ during the sampling period is 9.97±6.53 ug m$^{-3}$, the average concentration of $PM_{2.5}$ before rain is 11.13±7.21 ug m$^{-3}$, the average concentration of $PM_{2.5}$ after rain is 7.80±4.18 ug m$^{-3}$. The concentration of $PM_{2.5}$ is a significant drop. In addition, as shown in Table 1, there are differences in the correlation between $PM_{2.5}$ and $DTT_V$ before and after rain."

**Minor comments:**

More description on the light blocking and nitrogen environment system should be added? For example, how does the system look like? How to make sure it is sealed?

**Response**: *Thank you for your question. In the DTT experimental part, we used aluminum foil to wrap all the pipes to avoid light. Each line, reaction tube, pump A, pump B, and mixing tube were sealed with sealing plugs and sealing tape. During the experiment, the valve of $N_2$ was kept open, and the whole DTT experimental module was filled with $N_2$ through pump A and pump B. We add a new text to line 123 in the revised manuscript:*

"In the DTT reaction module, in order to avoid the influence of light and air on the experiment, all pipelines, reaction flasks and mixing flasks are sealed and protected from light by aluminum foil. The whole DTT experimental part was filled with $N_2$ by pump A and pump B before the experiment started."

What software was used to control the pumps, log data, etc?

**Response**: *The software to control the pump is "Serial Port Utility" and "Cadent connect". The data is recorded using "Spectra Suite", a software for measuring absorbance, which has the function of automatically saving data.*

"Then, use pump A to suck the mixed solution in the 1ml mixing bottle and transfer it to the reaction bottle to mix it with TCA" how much was withdrawn?

**Response**: "Then, use pump A to suck the mixed solution in the 1ml mixing bottle and transfer it to the reaction bottle to mix it with 1ml TCA"

Line 264-279, the discussion of diurnal variation and before and after rain is confusing. The authors are merely listing numbers without any discussion of what the comparison imply or suggest.

**Response**: *Thank you for your suggestion. The main purpose here is to express that there are obvious diurnal changes in ROS content, which may be related to different pollution emission sources. I added new content on line 279:*

"However, there are no obvious diurnal variation in $PM_{2.5}$ mass concentration. Therefore, the diurnal variation of DTT activity is assumed to be mainly attributed from different emission sources at the site."

Line 285, ng m-3. Typo? microg m-3?

**Response**: *Thanks for pointing it out, I have corrected it.*

"The average concentration of $PM_{2.5}$ during the sampling period is 9.97±6.53 ug m$^{-3}$, the average concentration of $PM_{2.5}$ before rain is 11.13±7.21 ug m$^{-3}$, the average concentration of $PM_{2.5}$ after rain is 7.80±4.18 ug m$^{-3}$. The concentration of $PM_{2.5}$ is a significant drop."

Technical comments:

There are many incomplete sentences and confusing use of language. The authors need to proofread the manuscript more carefully. Here are some examples:

Incomplete sentences:

"In order to more conveniently and accurately detect 15 the content of reactive oxygen in atmospheric particles hour by hour."

"In recent years, the online detection technology of ROS has been developed. However, there are few technical studies on online detection of ROS based on the DTT method."

"Clean the instrument 173 pipeline once a week, 5 times each time (Ultra-pure water)." A subject is missing. Many other sentences throughout the manuscript have the same problem. Please correct. Perhaps a passive tense is more appropriate.

"In the DTT experimental module, DTT and DTNB solutions are prepared every 4 days. Before each test, perform a comprehensive light and nitrogen bag inspection. To ensure the accuracy of the experimental data, a standard curve was measured before each experiment. The instrument pipeline is cleaned once a week, as shown in Figure 1. The programmable pump A and pump B are connected to the ultrapure water channel. During the cleaning process, all pipelines, reaction tubes and mixing tubes are cleaned."

"the average concentration of $PM_{2.5}$ after rain was 7.80±4.18 ng m-3, PM2.5 There is a significant drop in concentration."

"The average concentration of $PM_{2.5}$ during the sampling period is 9.97±6.53 ug m$^{-3}$, the average concentration of $PM_{2.5}$ before rain is 11.13±7.21 ug m$^{-3}$, the average concentration of $PM_{2.5}$ after rain is 7.80±4.18 ug m$^{-3}$. The concentration of $PM_{2.5}$ is a significant drop."

Confusing sentencences:

"the basis of the MARGA, which is a reliable field instrument…particle phases." MARGA is first mentioned here in the manuscript. What is field instrument? What is "transform the observation"? To what? How to transform?

**Response:** *Thanks for pointing it out, I have corrected it.*

"In addition, the present study is developed on the basis of the MARGA, which is a state-of-art instrument. MARGA measures near-real-time water-soluble particulate species and their gaseous precursors. (Chen et al., 2017)"

"we divided the DTTv daily activities" this sentence appears to be some sort of calculations but the authors meant to separate different days.

**Response:** *Thanks for pointing it out, I have corrected it.*

"To better understand the environmental factors affecting $DTT_V$, hourly data obtained by running the instrument is composited to obtain a diurnal profile of the DTT activity."

"The levels of these substances were not high during the sampling period and decreased to varying degrees after rain." How high is not high??

**Response:** *Thanks for pointing it out, I have corrected it.*

"The average concentration of $PM_{2.5}$ during the sampling period is 9.97±6.53 ug m$^{-3}$, the average concentration of $PM_{2.5}$ before rain is 11.13±7.21 ug m$^{-3}$, the average concentration of $PM_{2.5}$ after rain is 7.80±4.18 ug m$^{-3}$. The concentration of $PM_{2.5}$ is a significant drop. In addition, as shown in Table 1, there are differences in the correlation between $PM_{2.5}$ and $DTT_V$ before and after rain. Therefore, we suspect that the source of $DTT_V$ is different before and after the rain. BC and the polluting gases $SO_2$, $NO_x$, $NO_2$, CO, $Ca^{2+}$, $K^+$, $Mg^{2+}$ are often used as tracers of biomass burning, coal combustion, and dust storms. Compared with the early winter in the northern suburbs of Nanjing (Zhang et al., 2020), the levels of these substances decreased during the sampling period."

**Comment on amt-2021-407**
**Anonymous Referee #3**

The author's present a methodology where they incorporate a dithiothreitol (DTT) based assay into a previously established online method for particle composition analysis (MARGA). The authors make alterations compared to previous online DTT methods in the literature, including the use of nitrogen carrier gas and shielding from light in order to reduce the DTT background signal. The optimised DTT method is compared to current methods existing in the literature and deployed in ambient measurement campaigns where they correlate DTT activity with a range of inorganic ions, trace gasses and black carbon. However, there is a lack of technical detail in places, and the comparison between online and offline measurements requires additional clarification. I would recommend publication after addressing the following comments:

Line 111-113 - There is a lack of technical details in general in this section – is nitrogen continuously flowed through, if so what flow rate? How is the DTT reaction vial actually protected from light? As these are key modifications, there should be substantially more technical details added.

*Response*: *Thanks for the suggestion, Nitrogen does not flow continuously, we fill it with nitrogen before the experiment starts, and nitrogen is used to remove air from the instrument. We use aluminum foil to wrap the tubing and instruments to protect from light. We add a new text to line 124 in the revised manuscript:*

"In the DTT reaction module, to avoid the influence of light and air on the experiment, all pipelines, reaction flasks and mixing flasks are sealed and protected from light by aluminum foil. The whole DTT experimental part was filled with $N_2$ by pump A and pump B before the experiment started."

Line 172 – Additional details should be given regarding how offline PQN experiments were performed.

*Response*: *We add a new text to line 184 in the revised manuscript:*

"First, we select PQN with concentrations of 0.01, 0.02, 0.025, 0.05, 0.085 nmol $L^{-1}$ to compare online and offline DTT activity detection to determine the error of online and offline experiments. The details of PQN analysis can be found in Supplement S1."

Supplement S1:

"First, we configure PQN solutions with concentrations of 0.01, 0.02, 0.025, 0.05, 0.085 μM. Then, take 1.5 mL PQN solution and 5 mL 0.1 M potassium phosphate solution (adjust the pH to 7.4 after preparation) and mix in a 15 mL reaction flask. Next, add 0.5 mL of 1mM DTT to the reaction mixture, and place it in a constant temperature oscillator (THZ-D, Suzhou Peiying Experimental Equipment Co., Ltd.) at 37 °C and a rotation speed of 250 r/min. At the specified time interval (0, 10, 20, 30, 40 minutes), take out 0.5 mL of the reaction mixture and transfer it to another vial containing 0.5 mL of 10% w/v trichloroacetic acid (TCA) for termination reaction between DTT and sample solution. Then, add 50 μL of 1 mM DTNB (5,5'-dithiobis (2-nitrobenzoic acid)) to react with the remaining DTT in the solution. Finally, add 2 mL of 0.4 M Tris buffer (0.4 M Tris + 20 mM EDTA, adjust the pH to 8.9 after preparation), and use a

spectrophotometer to detect the absorbance at a wavelength of 412 nm, where the spectrophotometer includes an ultraviolet-visible (UV-VIS) light source (Ocean Optics DT-mini-2) and a multi-wavelength light detector (USB4000 micro fiber spectrometer), and the data acquisition software (Spectra Suite) to record the absorbance intensity at 412 and 700 nm (selected as the baseline absorbance of TNB)."

Line 174 – What are traditional samples, please specify.

*Response: Traditional sample trial production of real samples collected by high flow samplers. Here, it refers to the samples collected continuously for 24 hours using a large-flow $PM_{2.5}$ 188sampler (KC-6120) in Xuzhou. I have changed to offline samples.*

Line 184 – What are the $PM_{2.5}$ mass loadings on the filters collected for analysis?

**Response**: *The $PM_{2.5}$ mass loadings on the filters collected for analysis is between $110\mu g/m^3$ and $140\mu g/m^3$ per sampling film.*

Line 204 – Again, more detail required here regarding the nitrogen and light-reducing modifications.

**Response**: Thanks for the suggestion, *we add a new text to line 123 in the revised manuscript:*

"In the DTT reaction module, in order to avoid the influence of light and air on the experiment, all pipelines, reaction flasks and mixing flasks are sealed and protected from light by aluminum foil. The whole DTT experimental part was filled with $N_2$ by pump A and pump B before the experiment started."

Line 223 – If the slope is lower compared to Puthussery et al, does this not mean the response as a function of PQN concentrations is less, and thus the method is less sensitive to PQN? Is this slope corrected for baseline, accounting for background DTT consumption? This should be elaborated in more detail in the manuscript.

**Response**: *Thanks for the question, at pH 7.0, almost 100% of DTT was transformed to DTT-Disulfide by the catalyst 9,10-PQ (Li et al., 2009). This slope is corrected for the baseline to account for background DTT consumption. In addition, we think it is meaningless to compare the magnitude of the slope with Puthussery et al. Therefore, we delete all the comparisons about the slope size in the manuscript. We add a new text to line 240 in the revised manuscript:*

"At pH 7.0, almost 100% of DTT is transformed to DTT-Disulfide by the catalyst 9,10-PQ (Li et al., 2009). The analytical measurement part of the online DTT instrument is calibrated by measuring the DTT activity of PQN at different concentrations. As shown in Figure 4, the linear graph of DTT consumption rate and PQN concentration, which is after subtracting the blank DTT consumption rate."

Line 231 – How does this LOD compare to the other methods mentioned in the literature? And how does this compare to the LOD of the offline method used?

**Response**: *We add a new text to line 247 in the revised manuscript:*

"The limit of detection (LOD) of the system is defined as 3 times the standard deviation of the deionized water blank (N = 23), i.e., 0.024 nmol min$^{-1}$, which is significantly lower than the LOD of Puthussery et al. (0.24 nmol min$^{-1}$) and Fang et al. (0.31 nmol·min$^{-1}$)."

Line 254 – It is unclear to me what the purpose of the "PQN correction" is? Please elaborate.

**Response**: *When using PQN to compare online and offline, we found that the DTT consumption rate was deviated between online and offline conditions of the same concentration of PQN. We believe that it is caused by the experimental error of the online instrument. Therefore, when using the online instrument to measure the real sample, the experimental error is compensated by correction.*

Line 257 – Referring to the comparison of online and offline measurements in Figure 5. The authors state in the introduction that online methods are advantageous due to the online method capturing reactive components that offline methods currently do not, which is valid. Therefore, we would expect the online DTT signal to be higher than that of the offline method for an equivalent sample once normalised, due to the rapid capture of particles in the online method compared to offline. This is not the case in Figure 4 where the PQN slopes are the same, but the offline values are higher compared to online, why? Is this due to the optimisation of the online method? It is not clarified clearly. In Figure 5, the ambient samples measured offline have a higher DTT activity compared to online, and after the "PQN correction", the offline DTT activity is roughly equivalent to the online, if not still slightly higher for some samples. Puthuserry et al (2018), frequently cited in this manuscript, for instance show a higher online signal compared to offline. There is really limited description of the data in Figure 4 and Figure 5 in the manuscript, and as the online vs offline comparison is a key feature of implementing the DTT assay into an online methodology, this should be explained more clearly and in more detail.

**Response**: *We add a new text to line 269 in the revised manuscript:*

"The manual detection results are slightly higher than the automatic detection results, we assume that this is due to the instrument error caused by the complicated piping system of the online instrument.

As shown in Figure 6, the online and offline analysis of the DTT activity of 10 ambient particles, the slope (manual/automatic) obtained by orthogonal fitting is 1.14, the intercept is 0.19, and the correlation coefficient ($R^2$) is 0.954. We found that the real samples tested also had slightly higher offline results than online results. This is similar to our assumption."

Line 522 – Figure caption 1 is not sufficient to describe the method, please expand substantially to include more technical detail.

**Response**: *We add a new text to line 558 in the revised manuscript:*

[Figure]

Figure.2 Schematic diagram of DTT reaction part. (①-④ represents the DTT oxidation step,⑤-⑨ represents the DTT determination step. Blue indicates the ventilation line, all pipelines are wrapped in aluminum foil to protect from light.)

Technical corrections:

The manuscript could benefit from an additional proof reading for English as there are confusing and, in some cases, incomplete sentences throughout the manuscript.

Line 26 – reactive oxygen (species?) typo?
**Response**: *The full name of ROS is reactive oxygen species.*

Line 150 – deionized water (deionised water) typo.
**Response**: *Thanks for pointing it out, I have corrected it.*

Line 180 – the amount of what?
**Response**: *Thanks for pointing it out, we add a new text to line 193 in the revised manuscript:*

"The MARGA is calibrated using internal and external standards. The internal standard is a 10 mg L$^{-1}$LiBr solution. The external standard calibration is performed after replacing the anion and cation columns, and the replacement cycle is generally 4 to 5 months. At the same time, the MARGA system is cleaned with 1% hydrogen peroxide and 10% acetone solution, and the airflow is calibrated every two months. In the DTT experimental module, DTT and DTNB solutions are prepared every 4 days. Before each test, perform a comprehensive light and nitrogen bag inspection. To ensure the accuracy of the experimental data, a standard curve was measured before each experiment. The instrument pipeline is cleaned once a week, as shown in Figure 1. The programmable pump A and pump B are connected to the ultrapure water channel.

During the cleaning process, all pipelines, reaction tubes and mixing tubes are cleaned."

Line 294-296 – ng m-3, typo? ug m-3?

**Response**: *Thanks for pointing it out, we have revised the manuscript:*

"The average concentration of $PM_{2.5}$ during the sampling period is 9.97±6.53 ug $m^{-3}$, the average concentration of $PM_{2.5}$ before rain is 11.13±7.21 ug $m^{-3}$, the average concentration of $PM_{2.5}$ after rain is 7.80±4.18 ug $m^{-3}$. The concentration of $PM_{2.5}$ is a significant drop."

Line 526 – no x-axis title on Figure 2 (left), label both graphs (e.g. A and B)

**Response**: *Thanks for pointing it out, I have corrected it*

Line 533 – error given in Figure 3 but not in Figure 4 for the same data?

**Response**: *Figure 3 shows the rate of DTT consumption from 3 parallel experiments with different concentrations of PQN under offline conditions, so there is a standard deviation. Figure 4 shows the Comparison of the automated system with manual operation using PQN (9,10-phenanthraquinone), with no standard deviation.*